# Determination of Critical Limit of Zinc for Rice (*Oryza sativa* L.) and Potato (*Solanum tuberosum* L.) Cultivation in Floodplain Soils of Bangladesh

**Mamunur Rahman [1], Mohammad Mofizur Rahman Jahangir [1], Mohammad Golam Kibria [1], Mahmud Hossain [1], Md Hosenuzzaman [1], Zakaria M. Solaiman [2] and Md Anwarul Abedin [1,*]**

1   Department of Soil Science, Faculty of Agriculture, Bangladesh Agricultural University, Mymensingh 2202, Bangladesh; mamunur76@yahoo.com (M.R.); mmrjahangir@bau.edu.bd (M.M.R.J.); kibria.ss@bau.edu.bd (M.G.K.); mahmud.ss@bau.edu.bd (M.H.); hosen.ss@bau.edu.bd (M.H.)
2   UWA School of Agriculture and Environment, The University of Western Australia, Perth, WA 6009, Australia; zakaria.solaiman@uwa.edu.au
*   Correspondence: m.a.abedin@bau.edu.bd; Tel.: +880-1718031462

**Abstract:** The critical limit for zinc (Zn) varies from 0.38 to 2 μg/g soil depending on the crop and soil type. However, the critical limit for Zn was not explored recently for rice and potato cultivation in the floodplain soils of Bangladesh. A pot experiment was conducted to determine the critical limits of Zn in soil and plants for rice and potato cultivation in two agro-ecological zones (AEZs) of Bangladesh. The soil samples were collected from 20 different locations of Old Brahmaputra and Active Ganges Floodplains with low (<0.9 μg/g), medium (0.91–1.80 μg/g) and high (>1.80 μg/g) Zn status. The experiment was laid out in a factorial and completely randomized design with two levels of Zn ($Zn_0$ and $Zn_1$ (0 and 4.87 kg/ha as Zn sulphate)) applied to 20 different soil samples for rice and potato cultivation using three replications. The critical limit of Zn was determined through a graphical and statistical approach and crops were harvested at the stem elongation (for rice) and tuber filling (for potato) stage. The critical limit of Zn in soil for rice was found to be 0.8 and 0.85 μg/g by graphical and statistical methods, respectively, and both methods revealed the same value (0.73 μg/g) for potato. The critical limit of Zn in rice plants was 23.9 and 24.32 μg/g, whereas in potato plants it was 27.1 and 26.61 μg/g, determined by graphical and statistical methods, respectively. The added Zn supply in soil significantly increased the dry matter accumulation in rice (by 5.6%) and potato (by 10%) compared to no Zn supply. Therefore, a significant positive response to added Zn could be observed on crop growth and yield when the Zn concentration remained below the mentioned critical level for rice and potato cultivation in floodplain soils of Bangladesh.

**Keywords:** macronutrients; critical limits; floodplain soils; agro-ecological zones

## 1. Introduction

Zinc is an essential micronutrient for plant growth and development [1]. It performs a significant role in several plant metabolic processes, such as enzyme activity, chlorophyll formation, photosynthesis, respiration, the buildup of the cell wall and other biochemical functions [2,3]. Its importance in crop production is similar to that of other essential nutrients, as plants will not complete their life cycle without an adequate supply of Zn [4]. Zinc deficiency in cereal plants, including rice, is a severe problem that causes reduced agricultural productivity worldwide [1]. Zinc deficiency in crops reduces grain yield and hampers nutritional quality. Zinc deficiency was reported in the soils of lowland rice cultivation in Bangladesh, India, Pakistan, Philippines, Myanmar, Indonesia, Japan, Korea, Taiwan and Thailand [5]. Zinc was first reported as deficient in Bangladesh soil in 1982 [6]. Currently, an area of 2.76 million ha in Bangladesh has Zn deficiency in soils [7].

Rice (*Oryza sativa*) is one of the most important food crops and staple food for more than one-third of the world's population [8,9]. In Bangladesh, rice is the main component

of the daily diet. Considering cultivable land in Bangladesh, the most significant area is used for rice cultivation (11.68 million ha). It occupies the third position in the world for rice production (37.36 million MT), behind China and India. On the other hand, potato (*Solanum tuberosum*) is one of the most important crops cultivated globally [10] and the source of many essential compounds (starch, proteins, vitamins, sugars, minerals and other useful substances) in the human diet [11,12]. Potato, a staple food in many countries, is an important crop in Bangladesh, cultivated on 1.09 M ha. The adequate application of Zn is important along with all other essential nutrients to increase the growth and yield of potatoes. Thus, it is essential to identify specific nutrients in the soil below which the crop shows significant deficiency symptoms. The farmers will know the requirements of fertilizer application for a sustainable agricultural production system, especially for rice and potato.

Critical limit (CL) determines a threshold value of a nutrient in the soil, below which the crop will readily respond to its application. The critical limit of a nutrient in a plant refers to the level at or below which plants either develop deficiency symptoms, or there is a decrease in crop yields compared to optimum yields. The concept of CL of nutrients was introduced by Ulrich [13] and Smith [14]. However, the graphical method [15], and later the statistical approach [16], are being widely used to establish the CL of a nutrient. Importantly, there is no comparative report between these two approaches to determining CL in soil and plants in the floodplain soils of Bangladesh. Critical limits in soils and plants help in making practical recommendations to specific crops in a typical soil. Hence, the situation justifies the need to adjudicate and update the critical limit of different plant nutrients to formulate an optimum fertilizer dose for deficient nutrients in different crops and soils, in order to achieve a satisfactory crop yield. Therefore, it is essential to determine the CL for Zn in soil and plants to make Zn applications more efficient and rational for rice and potato cultivation in Bangladesh.

The present study focused on evaluating the CL for Zn in soil and plant for sustainable rice and potato cultivation in the floodplain soils of Bangladesh. Furthermore, this study will compare the graphical and statistical approaches to determining the CL for Zn in soil and plants for rice and potato predominantly growing in the floodplain soils of Bangladesh.

## 2. Materials and Methods

### 2.1. Soil Collection, Analysis and Test Crop

A total of 720 soil samples (topsoil: 0–15 cm soil depth) were collected from two agro-ecological zones (AEZs) of Bangladesh, representing floodplain soils (AEZ 9: Old Brahmaputra Floodplain and AEZ 9: Active Ganges Floodplain). The AEZs were developed mainly based on the soil characteristics and climatic conditions, and Bangladesh was divided into 30 AEZs, where rice and potato are predominantly grown in floodplain soils. All soils were air-dried, sieved ($\leq 2$ mm) and mixed until homogenous. The collected soil samples were analyzed to determine the different physico-chemical properties of soil such as texture, pH, electrical conductivity, organic matter and soil Zn status following standard methods [17]. The physico-chemical properties of the collected soil samples measured before starting the experiments are presented in Table 1.

Based on the Zn status in the collected soil samples, we selected 20 soil samples, comprising four high (>1.80 μg/g), four medium (0.91–1.80 μg/g) and 12 low (<0.9 μg/g) Zn-containing soils. These soils were used for conducting the pot experiments under glasshouse conditions. The non-draining plastic pots (4 kg capacity) were prepared with 3 kg of the collected soil samples for growing crops. Rice (var. BRRI dhan88) and potato (var. BARI Alu-7: Diamond) were used as test crops in this study.

**Table 1.** Selected soil properties of the collected soil samples before starting the experiment.

| Soil Sample No. | Textural Class (USDA) | pH | EC (dS/m) | Organic Matter (%) | Zn (µg/g) |
|---|---|---|---|---|---|
| 1 | Loam | 6.83 | 0.14 | 0.82 | 0.16 |
| 2 | Silt Clay Loam | 7.08 | 0.13 | 1.50 | 0.24 |
| 3 | Silt Loam | 6.89 | 0.24 | 1.10 | 0.28 |
| 4 | Silt Loam | 7.46 | 0.12 | 1.74 | 0.38 |
| 5 | Silt Loam | 5.51 | 0.07 | 1.91 | 0.40 |
| 6 | Loam | 5.67 | 0.11 | 1.5 | 0.46 |
| 7 | Sandy Loam | 7.25 | 0.11 | 1.34 | 0.52 |
| 8 | Silt Loam | 6.88 | 0.16 | 1.34 | 0.58 |
| 9 | Loam | 7.48 | 0.12 | 1.00 | 0.68 |
| 10 | Silt Loam | 5.26 | 0.07 | 2.86 | 0.70 |
| 11 | Silt Loam | 6.50 | 0.13 | 1.63 | 0.75 |
| 12 | Silt Clay Loam | 6.12 | 0.08 | 3.91 | 0.82 |
| 13 | Silt Loam | 5.78 | 0.07 | 2.59 | 0.92 |
| 14 | Silt Loam | 6.33 | 0.04 | 1.48 | 0.94 |
| 15 | Silt Loam | 5.25 | 0.06 | 2.86 | 1.34 |
| 16 | Loam | 6.00 | 0.08 | 1.62 | 1.35 |
| 17 | Clay loam | 5.20 | 0.10 | 3.81 | 2.24 |
| 18 | Silt Loam | 5.27 | 0.11 | 2.72 | 2.78 |
| 19 | Silt Loam | 7.34 | 0.16 | 1.60 | 2.81 |
| 20 | Silt Loam | 7.16 | 0.14 | 1.41 | 3.17 |
| Range | - | 5.20–7.48 | 0.04–0.24 | 0.82–3.91 | 0.16–3.17 |
| Mean | - | 6.36 | 0.11 | 1.94 | 1.08 |

## 2.2. Experimental Design and Approach

This study was conducted during winter (2019–2020) in the net-house of Department of Soil Science, Bangladesh Agricultural University, Bangladesh. The two-factorial experiment consisted of two levels of Zn for each plant species (0 and 4.87 kg/ha), rice and potato, and 20 selected soil samples collected from different AEZs with varying available Zn (ranging from 0.16 to 3.17 µg/g soil). The treatments were replicated three times using completely randomized design (six pots (2 Zn rates × 3 replication) for each soil and 120 pots (6 pots × 20 soils) for each crop). All soils were amended with the following basal nutrients (in mg/kg soil) mixed through the entire soil volume in each pot before sowing: N (150) from urea, P (25) from triple superphosphate, K (80) from muriate of potash and B (1) from boric acid. Analytical grade Zn sulphate ($ZnSO_4.2H_2O$) was used as the source of Zn for soil application.

This experiment was conducted in winter (November 2019 to February 2020) with an average day length of 11 h and temperature of 22 °C. Initially, four seedlings and a 45 g seed of potato was planted in each pot for rice and potato, respectively. After two weeks, rice seedlings were thinned by removing two seedlings from four. The pots were watered as per requirement; weeding and other intercultural operations were carried out as and when required. The plants were cut at the stem elongation stage for rice (76 days) and tuber filling stage for potato (45 days), washed with distilled water and dried in an oven at 65 °C for 48 h for recording dry biomass yield. Dried plant samples from each pot were grinded separately using a stainless still grinder. After that, the ground plant samples were digested in a mixture of 10:4:1 of $HNO_3:HClO_4:H_2SO_4$ on a hot plate and filtered by Whatman no. 42 to estimate Zn by atomic absorption spectrophotometer [18].

## 2.3. Critical Limit Determination by Graphical and Statistical Approaches

The graphical method of determining CL for Zn was followed as described by Cate and Nelson [15]. In this procedure, a scatter diagram of the relative yields (Bray's percent DM yield) as y-axis versus soil test values as x-axis was plotted. Bray's percent DM yield was determined via the following equation.

Bray's % dry matter (DM) yield = (DM yield without Zn treatment/DM yield with Zn treatment) × 100

The CL for Zn was also determined following a statistical approach as described by Cate and Nelson [16]. By this simple iterative process, a series of $R^2$ values could be

obtained for divisions made at various Zn levels. The critical level of Zn was determined from $R^2$ values where it was maximum.

### 2.4. Statistical Analysis

The raw data observed from pot experiments were analyzed using two factorial complete randomized design (CRD) to draw the valid differences among the treatments and soils. The data were subjected to two-way ANOVA and the significance of treatment on dry matter yield, concentration, and uptake of Zn by rice and potato plants was tested as described by Gomez and Gomez [19].

## 3. Results

### 3.1. Dry Matter Accumulation of Rice and Potato

The data of dry matter of plant biomass (g/pot) of rice plants grown at low-, medium- and high-Zn-status soils, as influenced by Zn application with $Zn_0$ and $Zn_{4.87}$, are presented in Table 2. The results demonstrated that the most significant, highest dry matter yield (g/pot) was found in high Zn status soils (26.53 g/pot) as compared to medium (20.33 g/pot) and low (14.96 g/pot) Zn status soils. The dry matter accumulation (g/pot) of rice plants was found to increase significantly due to the application of 4.87 kg $ZnSO_4.7H_2O$ per hectare (21.17 g/pot) over control (20.04 g/pot). The interaction effect between Zn status (low, medium and high) and treatment ($Zn_0$ and $Zn_{4.87}$) was non-significant with respect to the dry matter yield of rice plant.

**Table 2.** Effect of Zn application on dry matter yield (g/pot) of rice and potato on low-, medium- and high-Zn-containing soils in response to Zn amendment.

|  | **Dry Matter Yield of Rice (g/pot)** | **Dry Matter Yield of Potato (g/pot)** |
| --- | --- | --- |
| Zn Status | | |
| Low | 14.96 | 4.20 |
| Medium | 20.33 | 5.59 |
| High | 26.53 | 6.09 |
| SE $\pm$ | 0.61 | 0.16 |
| CD @ 5% | 1.21 | 0.31 |
| Treatment (T) | | |
| $Zn_0$ | 20.04 | 5.04 |
| $Zn_{4.87}$ | 21.17 | 5.55 |
| SE $\pm$ | 0.49 | 0.12 |
| CD @ 5% | 0.96 | 0.25 |
| Interaction (Zn $\times$ T) | | |
| SE $\pm$ | 0.7513 | 0.19 |
| CD @ 5% | NS | NS |

The interaction between Zn-status soils and levels of Zn on potato dry matter was found to be non-significant. Applying Zn significantly increased the dry matter yield in potatoes in most of the cases of this experiment (Table 2). The addition of Zn (4.87 kg/ha) increased dry matter yield (5.55 g/pot) compared to control (5.04 g/pot), as shown in Table 2. The high-Zn-status soils obtained a significantly higher dry matter yield (6.09 g/pot), followed by medium-Zn-status (5.59 g/pot).

### 3.2. Critical Limits of Zn in Soil and Plant for Rice

The critical limit of available Zn (DTPA extractable) for rice in soil was found to be 0.85 µg/g by a statistical method (Table S1) and 0.8 µg/g by a graphical method (Figure 1).

The Zn available in the soil (DPTA extractable) was arranged in ascending order (P1) and then Bray's percent was reported to the respective available Zn status of the soil. The mean relative yield in P1, correlated sum of squares of deviation from the mean of P2 (CSS-I), the mean relative yield in P2, and the correlated sum of squares of deviation from mean of P2 (CSS-II) were calculated. From the highest $R^2$ (0.871) the postulated critical limit of Zn for rice was found as 0.85 µg/g of Zn available in the soil.

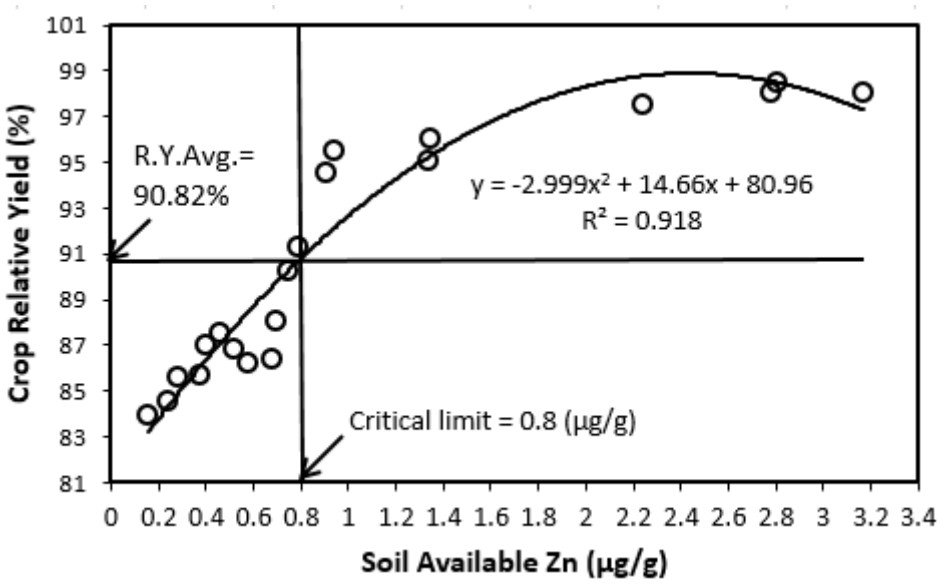

**Figure 1.** Scatter diagram of available soil Zn vs. Bray's percent dry matter yield of rice.

The critical limit of Zn for rice in plants was estimated by arranging the Zn concentration in ascending order. Therefore, the Bray's percent dry matter yield was reported for the respective values of Zn concentration in rice plants. The values of the mean relative yield in P1, corrected sum of squares of deviations from mean of P1 (CSS-I), mean relative yield in P2 and corrected sum of squares of deviations from mean P2 (CSS-II) were calculated. From these data, $R^2$ values were calculated for every concentration of Zn in rice plants. The postulated critical limit of Zn in rice plants was found to be 24.32 and 23.9 µg/g by graphical (Figure 2) and statistical methods (Table S2).

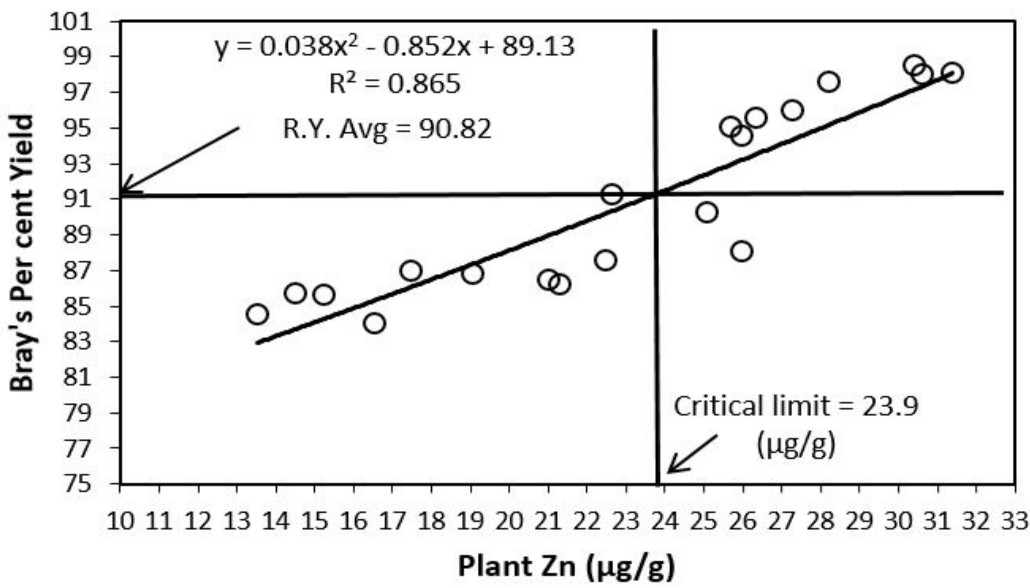

**Figure 2.** Scatter diagram of Zn concentration in plant vs. Bray's percent dry matter yield of rice.

### 3.3. Critical Limits of Zn in Soil and Plant for Potato

The data of dry matter of plant biomass (g/pot) and Bray's percent yield of potato were utilized for evaluating the critical limit of soil and plant Zn for potato (Tables S3 and S4). The scatter diagram with soil-available Zn vs. Bray's percent yield in potato are depicted in Figure 3 for the graphical method and the postulated critical limit shown in Table S3 for the statistical method is 7.3 µg/g, indicating the critical limit of Zn in soil, respectively.

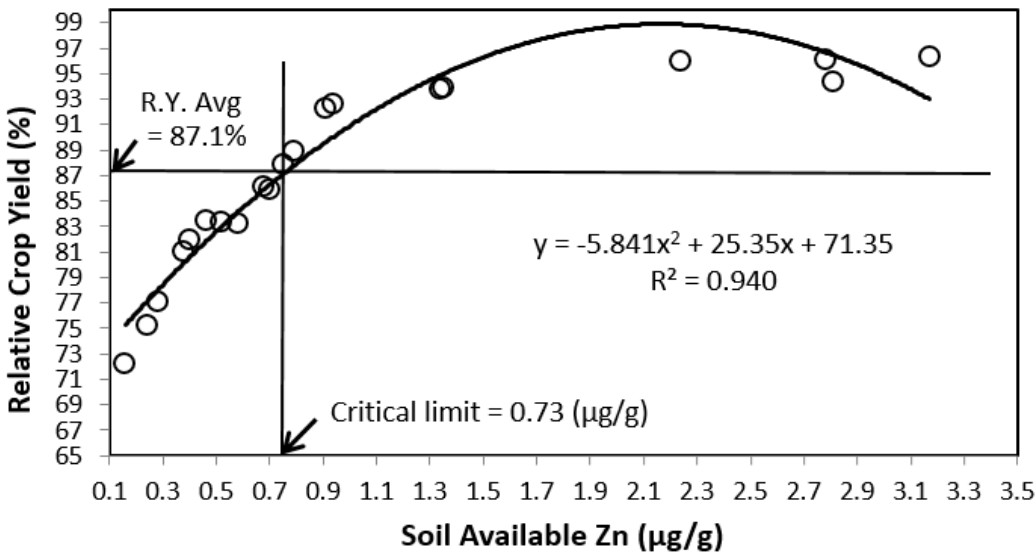

**Figure 3.** Scatter diagram of available soil Zn vs. Bray's percent dry matter yield of potato.

The data showed that the critical limits of Zn for potatoes were 26.61 and 27.1 µg/g by statistical and graphical method (Table S4 and Figure 4), respectively. The data suggest that if the plant contains less than 27.1 µg/g Zn, the crop would respond to application of Zn, and this critical limit is important for predicting the sufficiency and deficiency level of Zn in plants.

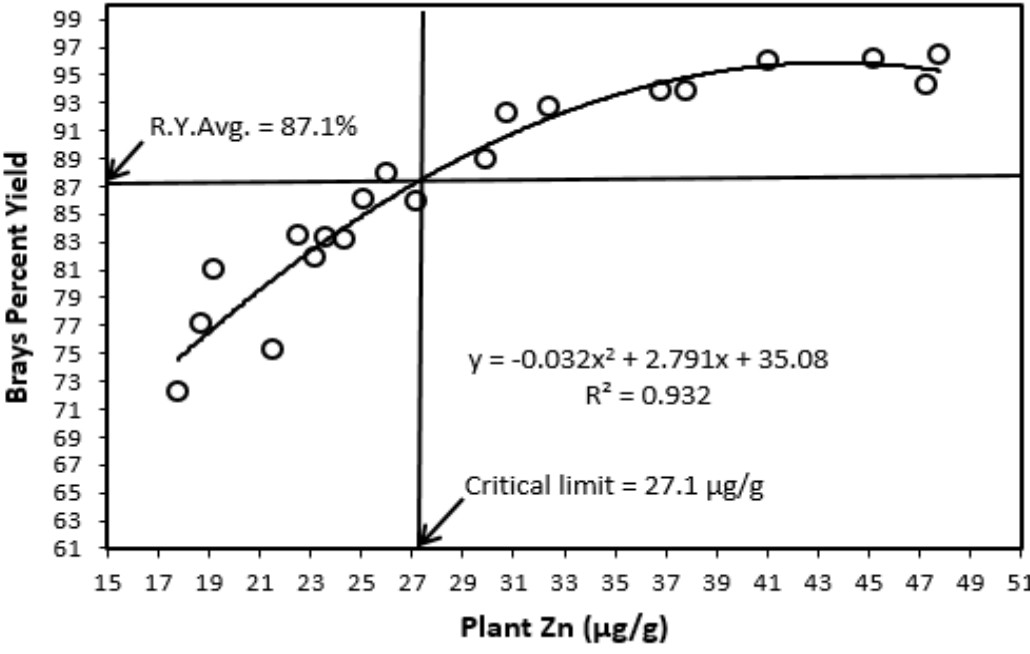

**Figure 4.** Scatter diagram of Zn concentration in plant vs. Bray's percent dry matter yield of potato.

## 4. Discussion

A set of 20 representative soils were selected from 720 soils throughout Bangladesh covering two AEZs (AEZ-9 and AEZ-10), having a wide range in texture, general soil type, cropping pattern and land type. The soils were dominated by rice-based cropping patterns and belong to subtropical monsoon climates with a wide variation in rainfall patterns, temperature and humidity. Rice is predominately grown in AEZ-9 and potatoes in AEZ-10 [7]. The soil analysis revealed that the soil's Zn concentration and other physicochemical parameters varied significantly, since the soil samples were collected from different parts of Bangladesh. The varied critical value of Zn for rice and potato [7] could be due to the diversified sample approach. Murthy [20], who also highlighted that a change in sample sites causes significant variation in nutrient CL, corroborated this idea.

Based on this present study, the CL for Zn in soil for rice and potato increased compared to the previous values. The critical limit of available Zn for rice in the present study was 0.80 µg/g and 0.85 µg/g in graphical and statistical methods, respectively. For potato, it was 0.73 µg/g in both methods. These values are close to a critical level for Zn, as observed by other authors [21–25]. The previously observed CL of Zn for rice and potato was 0.6 µg/g, which is lower than the present study findings. It is expected that rice crops will respond to Zn application when the soils contain less than 0.80 µg/g and 0.85 µg/g Zn content, and the potato will respond when the Zn concentration in soil is 0.73 µg/g. This increase in CL for Zn might occur due to the intensive cropping system, limited application of Zn fertilizer and high soil pH, favoring enhanced Zn adsorption and precipitation in soil [26]. Instead of using soil test data, the CL for Zn in plant tissues can also be utilized as an indicator to assess the requirement of Zn application. Based on the present study, the tissue Zn concentration was 23.9 µg/g and 24.32 µg/g for rice and 27.1 µg/g and 26.61 µg/g for potato by graphical and statistical methods, which is also in agreement with Dudde and Maleware [27]. Thus, anyone can determine the requirement of Zn by assessing the CL both for soil and plant tissue concentrations.

The CL of a nutrient in soil may vary depending on crops, soil, and extraction methods. Zn availability in soil depends on the soil type (Zn deficiency occurs most often in sandy soil), soil pH (available at a low soil pH) and organic matter content in soils. Thus, it is essential to determine and update the CL for Zn to formulate an appropriate dose of deficient nutrients for various crops and soils. Due to current agricultural practices, the CL will continue to change, and constant monitoring is required to maximize fertilizer use efficiency and maintain a sustainable production system. Due to the available soil Zn corresponding to a corrected sum of squares for the population in this study to determine the predictability value, a lower value in the statistical technique might arise ($R^2$). However, this dispute requires further investigation to know why the CL is lower in the statistical method than the graphical method. It was already reported that the statistical approach for determining CL provides a lower value than the graphical approach [28]. Researchers can use any method to determine the CL of nutrients in the soil because the CL values are similar in both methods. Because a more significant percentage of soils will fail to comply with the CL, we propose utilizing the maximum critical value calculated from the graphical and statistical approach to assure increased crop output. A competent approach should be able to forecast the amount of plant-available nutrients and the fertilizer responsiveness of crops grown in a variety of soils. In this regard, determining CL via two different methodologies is imperative in determining the optimum fertilizer demand [28].

Applying Zn to soil significantly improved the dry shoot biomass of crops cultivated in various soils. The increase was higher in low-Zn-containing soils than high-Zn-containing soils. The application of Zn to soil exhibited approximately a 10% increase in dry shoot biomass compared to when no Zn was applied to soil [29]. Naik and Das [30] reported that the application of Zn to low land rice soil resulted in the 37.8% and 20.9%, the most significant increase in the grain and straw yield of rice, respectively, over the control. Soils with a low Zn content gave a maximum response to tuber yield and increased potato yield by 8% [31]. Dry matter accumulation is a crucial crop growth metric that is often used

to calculate the economic returns influenced by the effects of various treatments. In crop ecosystems, Zn is frequently a limiting element for shoot biomass yield [32]. The efficient uptake and metabolism of available Zn might contribute to the gains in shoot biomass output seen in this study, which is also in agreement with Rashid and Fox [33]. Thus, applying Zn to the soil at or above the CL will aid in crop development and productivity in various soil types.

## 5. Conclusions

We conclude that Zn significantly increased shoot dry matter yield in rice and potato. Overall, the additional Zn supply should provide a yield benefit in a soil with a Zn content of less than 1 μg/g. To be specific, the CL of available Zn for soil in rice was 0.8 μg/g and 0.85 μg/g in graphical and statistical methods, respectively, below which there is a strong probability of observing a successful response to the added Zn fertilizer. In both methods, the CL for potato was 0.73 μg/g, which was higher than the present CL (0.6 μg/g soil). The critical plant tissue concentration of Zn was 23.9 μg/g and 24.32 μg/g for rice, and 27.1 μg/g and 26.61 μg/g for potato by graphical and statistical methods, respectively. The findings from this study can be used for updating fertilizer recommendation guides for efficient fertilizer applications in Bangladesh. This study may also ensure that Zn fertilization is crucial for achieving a higher economic yield of rice and potato and sustainable soil Zn management below the specified level. However, future research should confirm the current study's findings in actual field conditions.

**Supplementary Materials:** The following are available online at https://www.mdpi.com/article/10.3390/su14010167/s1, Table S1: Critical limit of soil available Zn for rice by statistical method; Table S2: Critical limit of plant Zn for rice by statistical method; Table S3: Critical limit of soil available Zn for potato by statistical method; Table S4: Critical limit of plant Zn for potato by statistical method.

**Author Contributions:** Conceptualization and experimental design, M.A.A., M.M.R.J. and M.H. (Mahmud Hossain); research and data collection, M.R.; data analysis, M.A.A. and M.R.; writing— original draft, M.R.; writing—reviewing and editing, M.G.K., M.H. (Md Hosenuzzaman) and Z.M.S.; supervision, M.A.A. All authors have read and agreed to the published version of the manuscript.

**Funding:** The authors acknowledge the financial support from the World-Bank-funded NATP-Phase II (Project ID: P149553) and the Government of Bangladesh.

**Institutional Review Board Statement:** Not applicable.

**Informed Consent Statement:** Not applicable.

**Data Availability Statement:** The data are available from the corresponding author upon reasonable request.

**Acknowledgments:** The first author gratefully acknowledges the financial support from the World Bank and the logistic support from the Department of Soil Science, Bangladesh Agricultural University in collecting soil and conducting experiments.

**Conflicts of Interest:** The authors declare no conflict of interest.

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
