# Peer review of "Determination of Critical Limit of Zinc for Rice (Oryza sativa L.) and Potato (Solanum tuberosum L.) Cultivation in Floodplain Soils of Bangladesh"

_sustainability, doi:10.3390/su14010167_

Round 1

Reviewer 1 Report

This paper addresses the potential benefits of additional Zn added to low-Zn soils for cultivation of rice and potato in Bangladesh by determining the critical limit needed, as well as assessing positive effects of Zn addition across a wide range of soils with differing Zn content. Although thus is a pot experiment, a diverse range of different soils from the region were collected and used, representing Zn content ranging from very low to high. The experiment is well-laid out, with appropriate replication, and the methodologies are adequate.  However, the overall experiment was not repeated, which is a weakness of this research.  The data collected is quite useful and informative. However, the presentation of data needs much re-focus and revision. Yield data should be presented more fully, in that yield both with and without added Zn must be presented for each Zn status group, not just for overall, as it is in the specific groups (low, Med, high) that the most important data and effects are observed, and should be the focus of the paper. Text needs to more fully present these data. Tables 3-6, contain much redundant and/or unneeded information (calculations used for conducting critical limits) and should be reduced and combined into one table for each crop (containing both soil and plant Zn data). Overall, there is too much emphasis placed on determining, presenting, and discussing the exact calculation of the critical limits, when the primary focus should be on the effect of the Zn addition on yield in these soils (and primarily the Zn-deficient soils), as this is what proves the thesis of the paper. In the end, the primary conclusion should be that Zn additions should provide a yield benefit in any soil with a Zn content of less than ~1 ug/g (the exact CL is not that important, as long as above a general threshold). 

There are also numerous awkward wordings and phrasings throughout (use of articles, prepositions, pronouns, etc.). I have made some suggested revisions, but entire manuscript needs to be checked for these types of English grammar issues. Additional suggestion: the phrase should be 'CL for Zn', rather than 'CL of Zn' throughout paper. 

Overall, it provides useful information and data. However, some major revision is needed in re-organizing the presentation of data and results, and re-focusing discussion to highlight the most important and significant findings and conclusions, as well as revising the English grammar. I have made numerous additional notes and comments throughout the attached edited pdf file. 

Author Response

Please find the attached author response 

Reviewer 2 Report

Dear authors

The authors measured the Zn content in the shoot dry matter to establish the required minimum and maximum quantities of Zn in rice and potato. This manuscript is unable to be accepted for the following reasons:

  1. I believe that the experiment's design is inadequate since, as shown in table 1, there is a wide range of variation in EC, pH, and organic matter in the 20 soil samples obtained. As a result, categorizing soil samples into three groups based on Zn content is illogical and incorrect because the soil categories are not homogeneous in terms of EC, pH, and organic matter. As shown in Table 1, there is no relationship between soil Zn content and other soil factors such as EC, pH, and organic matter. Furthermore, the authors do not display or assess the content of macro-elements in each soil sample, despite the fact that all of these soil properties and constituents have a significant impact on the formation of dry matter. Because there is a substantial variance in organic matter and EC between soil status, the authors may see these non-homogenized in soil sample categories in organic matter and EC in table 2. As a result, the classification of 20 soil samples based on Zn concentration does not result in three mixtures of soil homogenizing in soil properties.
  2. The use of Zn had a minor impact on the formation of dry materials. As a result, in terms of economics, this experiment is unimportant in the agriculture sector.
  3. The authors measured only a few characteristics. They just measured the dry matter of the shoots. Some traits related to growth and yield in rice and potato plants should be measured by the authors. Furthermore, the authors should conduct certain physiological and biochemical parameters because the application of Zn has a significant impact on the physiological and biochemical processes in plants.
  4. The authors should test the Zn content in rice seeds and potato tubers because a significant accumulation of Zn in rice seeds or potato tubers is regarded a toxic situation.
  5. The authors stated on page 1 of L14 that the critical limit of Zn for rice and potato cultivation 14 in floodplain soils of Bangladesh had not been investigated recently. That, in my opinion, is incorrect because there have been various studies conducted on the critical limit of Zn in rice or potato, as follows:
  • Assessment of critical limit of zinc for rice, groundnut and potato in red and laterite soils of Odisha (2016) in Oryza Journal by DR Sarangi, D Jena and AK Chatterjee
  • Critical limits of zinc in soil and rice plant grown in alluvial soils of West Bengal, India (2014) in SAARC Journal of Agriculture Journal by Mahata, P. Debnath, S. Ghosh
  1. I also made some notes about manuscript organization and writing:
  • All sections (abstract, introduction, etc.) are easily written, and several lines are repeated.
  • The discussion is too weak, and the majority of them are about the reputation of the findings.
  • The quality of the figures and tables is poor.
  • The methodology is not written in an understandable manner.
  • The two factors, as well as the level of each factor, are not well defined.
  • The mean comparison method is not mentioned.
  • The table's title does not completely correlate to its content.
  • The manuscript's English language level needs to be improved.

Best regards

Author Response

(The authors gave the same response as above.)

Reviewer 3 Report

This manuscript is interesting to critical limit of zinc for agricultural crops. However, there are a number of issues in current form which requires revision.

The tables themselves should be streamlined to highlight the key concepts, and then the data-heavy tables (Table 3, 4, 5 and 6) are more appropriate as an appendix.  The result section also should be streamlined. The authors would do well to refer to other peer-reviewed publications for guidelines on what is most appropriate in tables, results, and figures, and what is better placed in an appendix.

The analysis was difficult for me to understand. I could only grasp a few of the basic methods of the analysis by looking at the figures and tables themselves.  A more minor problem was that the paper was difficult to follow, I believe mostly due to problems with English language. The discussion especially needs to be reviewed and edited. I did not include comments on the discussion to the authors because I think need more statistics for publication than this.

Overall, I suggest the authors work on improving the analysis with more modern and robust methods, streamlining the results and tables, and getting assistance in writing the methods and discussion more clearly.  I think the data themselves are original, valuable, and should be published after some of these items are addressed by the authors.

Author Response

(The authors gave the same response as above.)

Round 2

Reviewer 1 Report

The authors have adequately addressed the minor grammatical and awkward wording issues in this revised version of the manuscript. Thank you for that. It reads better and more clearly now. Unfortunately, however, the authors have chosen to not address the most important and critical issue with the paper, regarding the presentation of their data. It is essential that the yield data is presented for both with and without Zn addition for each Zn soil status group, showing the differences among the groups in response to Zn. The author's response that they chose not to present this data because they are focusing on the CL calculations is entirely inadequate and inappropriate. This is the central experiment of this study, and the results need to be clearly and accurately presented here, and that has not been done. The way the authors present the yield data, as only the overall average effect over all soils misrepresents the actual results due to the variable response of the different levels of Zn status, thus presenting a misleading result and conclusion, which actually undermines the main point of the paper. Due to these different responses based on Zn status, the overall response, averaged across all soils does not really represent or apply to any of the actual soils sampled (in that the overall increase of 5.6% for rice and 10% for potato is not even close to accurate for either the low [which is much higher response] or the high [much lower response] groups, and thus presents very misleading information. This is not something that the authors can just say that they don't want to include, it is an essential component of their experiment, and they need to clearly and accurately present and discuss it. This was the whole point of dividing the soils into these groups of low, medium, and high Zn status. It also is the most important and relevant data contained in the entire paper, so it is inconceivable that the authors would just say that they choose not to include it. It is essential, and the paper does not warrant publication without it. Table 1 needs to be changed to include this full presentation of the yield data (for each Zn status group, with and without Zn addition), as well as presentation in the Results section, and appropriate discussion and conclusions as well. 

Another recommendation that was not accepted was the content of Tables 3-6.  As I indicated, these tables are filled with much redundant and unneeded information for the calculation of CL. The authors have simply moved the tables to be Supplementary materials rather than part of the main paper in response to another reviewer's comments. however, this does not correct or improve the unnecessary and redundant info throughout the tables, as they still are exactly the same. As I recommended, the 4 tables still need to be reduced down to 2, one for rice, one for potato, and include data for both soil and plants (and reduce the redundant and unneeded columns). Even as Supplementary tables, they still need to be improved and reduced.

Thus, even though the authors have revised and improved the paper to some degree, virtually all of my previous comments (other than grammar and wording issues) are still relevant and still apply to the paper, and unfortunately, this paper will not be ready for publication until these issues are addressed.   

Author Response

We have improved the figures in the revised manuscript

Reviewer 2 Report

Dear Authors

Your second response does not sit well with me. You are correct that there is a considerable difference, yet it is a minor change. Concerning the comparison method between means, you said Duncan test, which is incorrect since each pair of means has a Duncan value, but in your results, there is one value of comparison for all pair of means, so I believe the comparison method in your results is LSD, please verify it.

best wishes

Author Response

Please find the attached response 
